# BAYESIAN DEEP LEARNING VIA STOCHASTIC GRADIENT MCMC WITH A STOCHASTIC APPROXIMATION ADAPTATION

## ABSTRACT

We propose a robust Bayesian deep learning algorithm to infer complex posteriors with latent variables. Inspired by dropout, a popular tool for regularization and model ensemble, we assign sparse priors to the weights in deep neural networks (DNN) in order to achieve automatic "dropout" and avoid over-fitting. By alternatively sampling from posterior distribution through stochastic gradient Markov Chain Monte Carlo (SG-MCMC) and optimizing latent variables via stochastic approximation (SA), the trajectory of the target weights is proved to converge to the true posterior distribution conditioned on optimal latent variables. This ensures a stronger regularization on the over-fitted parameter space and more accurate uncertainty quantification on the decisive variables. Simulations from large-p-small-n regressions showcase the robustness of this method when applied to models with latent variables. Additionally, its application on convolutional neural networks (CNN) leads to state-of-the-art performance on MNIST and Fashion MNIST datasets and improved resistance to adversarial attacks.

## 1 INTRODUCTION

Bayesian deep learning, which evolved from Bayesian neural networks (Neal, 1996; Denker and leCun, 1990), provides an alternative to point estimation due to its close connection to both Bayesian probability theory and cutting-edge deep learning models. It has been shown of the merit to quantify uncertainty (Gal and Ghahramani, 2016b), which not only increases the predictive power of DNN, but also further provides a more robust estimation to enhance AI safety. Particularly, Gal and Ghahramani (2016a;b) described dropout (Srivastava et al., 2014) as a variational Bayesian approximation. Through enabling dropout in the testing period, the randomly dropped neurons generate some amount of uncertainty with almost no added cost. However, the dropout Bayesian approximation is variational inference (VI) based thus it is vulnerable to underestimating uncertainty.

MCMC, known for its asymptotically accurate posterior inference, has not been fully investigated in DNN due to its unscalability in dealing with big data and large models. Stochastic gradient Langevin dynamics (SGLD) (Welling and Teh, 2011), the first SG-MCMC algorithm, tackled this issue by adding noise to a standard stochastic gradient optimization, smoothing the transition between optimization and sampling. Considering the pathological curvature that causes the SGLD methods inefficient in DNN models, Li et al. (2016) proposed combining adaptive preconditioners with SGLD (pSGLD) to adapt to the local geometry and obtained state-of-the-art performance on MNIST dataset. To avoid SGLD's random-walk behavior, Chen et al. (2014) proposed using stochastic gradient Hamiltonian Monte Carlo (SGHMC), a second-order Langevin dynamics with a large friction term, which was shown to have lower autocorrelation time and faster convergence (Chen et al., 2015). Saatci and Wilson (2017) used SGHMC with GANs (Goodfellow et al., 2014a) to achieve a fully probabilistic inference and showed the Bayesian GAN model with only 100 labeled images was able to achieve 99.3% testing accuracy in MNIST dataset. Raginsky et al. (2017); Zhang et al. (2017); Xu et al. (2018) provided theoretical interpretations of SGLD from the perspective of non-convex optimization, echoing the empirical fact that SGLD works well in practice.

When the number of predictors exceeds the number of observations, applying the spike-and-slab priors is particularly powerful and efficient to avoid over-fitting by assigning less probability mass on

the over-fitted parameter space (Song and Liang, 2017). However, the inclusion of latent variables from the introduction of the spike-and-slab priors on Bayesian DNN is not handled appropriately by the existing methods. It is useful to devise a class of SG-MCMC algorithms that can explore the Bayesian posterior as well as optimize the hidden structures efficiently.

In this paper, we propose a robust Bayesian learning algorithm: SG-MCMC with a stochastic approximation adaptation (SG-MCMC-SA) to infer complex posteriors with latent variables and apply this method to supervised deep learning. This algorithm has four main contributions: 1. It enables SG-MCMC method to efficiently sample from complex DNN posteriors with latent variables and is proved to converge; 2. By automatically searching the over-fitted parameter space to add more penalties, the proposed method quantifies the posterior distributions of the decisive variables more accurately; 3. The proposed algorithm is robust to various hyperparameters; 4. This method demonstrates more resistance to over-fitting in simulations, leads to state-of-the-art performance and shows more robustness over the traditional SG-MCMC methods on the real data.

## 2   STOCHASTIC GRADIENT MCMC

We denote the decaying learning rate at time $k$ by $\epsilon^{(k)}$, the entire data by $\mathcal{D} = \{d_i\}_{i=1}^N$, where $d_i = (x_i, y_i)$, the log of posterior by $L(\beta)$, $\nabla$ as the gradient of any function in terms of $\beta$. The mini-batch of data $\mathcal{B}$ is of size $n$ with indices $\mathcal{S} = \{s_1, s_2, ..., s_n\}$, where $s_i \in \{1, 2, ..., N\}$. Stochastic gradient $\nabla \tilde{L}(\beta)$ from a mini-batch of data $\mathcal{B}$ randomly sampled from $\mathcal{D}$ is used to approximate the true gradient $\nabla L(\beta)$:

$$\nabla \tilde{L}(\beta) = \nabla \log \mathrm{P}(\beta) + \frac{N}{n} \sum_{i \in \mathcal{S}} \nabla \log \mathrm{P}(d_i | \beta).$$

The stochastic gradient Langevin dynamics (no momentum) is formed as follows:

$$\beta^{(k+1)} = \beta^{(k)} + \epsilon^{(k)} G \nabla \tilde{L}(\beta^{(k)}) + \mathcal{N}(0, 2\epsilon^{(k)} G \tau^{-1}), \tag{1}$$

where $\tau > 0$ denotes the temperature, $G$ is a positive definite matrix to precondition the dynamics. It has been shown that SGLD asymptotically converges to a stationary distribution $\pi(\beta | \mathcal{D}) \propto e^{\tau L(\beta)}$ (Teh et al., 2015; Zhang et al., 2017). As $\tau$ increases, the algorithm tends towards optimization with underestimated uncertainty. Another variant of SG-MCMC, SGHMC (Chen et al., 2014; Ma et al., 2015), proposes to use second-order Langevin dynamics to generate samples:

$$\begin{cases} \beta^{(k+1)} = \beta^{(k)} + \epsilon^{(k)} M^{-1} r^{(k)} \\ \\ r^{(k+1)} = r^{(k)} + \epsilon^{(k)} \nabla \tilde{L}(\beta^{(k)}) - \epsilon^{(k)} C M^{-1} r + \mathcal{N}(0, \epsilon^{(k)}(2C - \epsilon^{(k)} \hat{B}^{(k)}) \tau^{-1}) \end{cases} \tag{2}$$

where $r$ is the momentum item, $M$ is the mass, $\hat{B}$ is an estimate of the error variance from the stochastic gradient, $C$ is a user-specified friction term to counteracts the noisy gradient.

## 3   STOCHASTIC GRADIENT MCMC WITH A STOCHASTIC APPROXIMATION ADAPTATION

Dropout has been proven successful, as it alleviates over-fitting and provides an efficient way of making bagging practical for ensembles of countless sub-networks. Dropout can be interpreted as assigning the Gaussian mixture priors on the neurons (Gal and Ghahramani, 2016c). To mimic Dropout in our Bayesian CNN models, we assign the spike-and-slab priors on the most fat-tailed weights in FC1 (Fig. 1). From the Bayesian perspective, **the proposed robust algorithm distinguishes itself from the dropout approach in treating the priors: our algorithm keeps updating the priors during posterior inference, rather than fix it**. The inclusion of scaled mixture priors in deep learning models were also studied in Blundell et al. (2015); Li et al. (2016) with encouraging results. However, to the best of our knowledge, none of the existing SG-MCMC methods could deal with complex posterior with latent variables. Intuitively, the Bayesian formulation with model averaging and the spike-and-slab priors is expected to obtain better predictive performance through averages from a "selective" group of "good" submodels, rather than averaging exponentially many posterior probabilities roughly. For the weight priors of the rest layers (dimension $u$), we just assume they follow the standard Gaussian distribution, while the biases follow improper uniform priors.

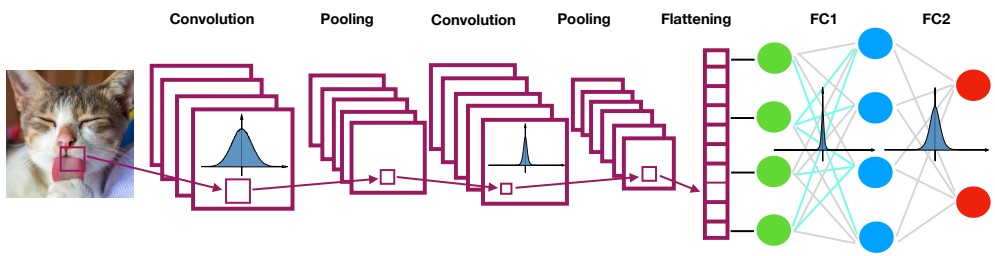

Figure 1: Model Structure

## 3.1 CONJUGATE SPIKE-AND-SLAB PRIOR FORMULATION FOR DNN

Similarly to the hierarchical prior in the EM approach to variable selection (EMVS) (Rořková and George, 2014), we assume the weights $\boldsymbol{\beta} \in \mathbb{R}^p$ in FC1 follow the spike-and-slab mixture prior

$$\pi(\boldsymbol{\beta}|\sigma^2, \boldsymbol{\gamma}) = \mathcal{N}_p(\mathbf{0}, \mathbf{V}\sigma, \boldsymbol{\gamma}), \tag{3}$$

where $\boldsymbol{\gamma} \in \{0, 1\}^p$, $\sigma \in \mathbb{R}$, $\boldsymbol{V}_{\sigma, \boldsymbol{\gamma}} = \sigma^2 \operatorname{diag}(a_1, \ldots, a_p)$ with $a_j = (1 - \gamma_j)v_0 + \gamma_j v_1$ for each $j$ and $0 < v_0 < v_1$. By introducing the latent variable $\gamma_j = 0$ or 1, the mixture prior is represented as

$$\beta_j|\gamma_j \sim (1 - \gamma_j)\mathcal{N}(0, \sigma^2 v_0) + \gamma_j \mathcal{N}(0, \sigma^2 v_1). \tag{4}$$

The interpretation is: if $\gamma_j = 0$, then $\beta_j$ is close to 0; if $\gamma_j = 1$, the effect of $\beta_j$ on the model is intuitively large. The likelihood of this model given a mini-batch of data $\{(\boldsymbol{x}_i, y_i)\}_{i \in \mathcal{S}}$ is

$$\pi(\mathcal{B}|\boldsymbol{\beta}, \sigma^2) = \begin{cases} (2\pi\sigma^2)^{-n/2} \exp\left\{-\dfrac{1}{2\sigma^2} \sum_{i \in \mathcal{S}}(y_i - \psi(\boldsymbol{x}_i; \boldsymbol{\beta}))^2\right\} & \text{(regression)} \\[2ex] \prod_{i \in \mathcal{S}} \dfrac{\exp\{\psi_{y_i}(\boldsymbol{x}_i; \boldsymbol{\beta})\}}{\sum_{t=1}^K \exp\{\psi_t(\boldsymbol{x}_i; \boldsymbol{\beta})\}} & \text{(classification)}, \end{cases} \tag{5}$$

where $\psi(\boldsymbol{x}_i; \boldsymbol{\beta})$ can be a mapping for logistic regression or linear regression, or a mapping based on a series of nonlinearities and affine transformations in the deep neural network. In the classification formulation, $y_i \in \{1, \ldots, K\}$ is the response value of the $i$-th example.

In addition, the variance $\sigma^2$ follows an inverse gamma prior

$$\pi(\sigma^2|\boldsymbol{\gamma}) = IG(\nu/2, \nu\lambda/2). \tag{6}$$

The i.i.d. Bernoulli prior is used since there is no structural information in the same layer.

$$\pi(\boldsymbol{\gamma}|\delta) = \delta^{|\boldsymbol{\gamma}|}(1 - \delta)^{p - |\boldsymbol{\gamma}|}, \text{ where } \pi(\delta) \propto \delta^{a-1}(1 - \delta)^{b-1} \text{ and } \delta \in \mathbb{R}. \tag{7}$$

Finally, our posterior density follows

$$\pi(\boldsymbol{\beta}, \sigma^2, \delta, \boldsymbol{\gamma}|\mathcal{B}) \propto \pi(\mathcal{B}|\boldsymbol{\beta}, \sigma^2)^{\frac{N}{n}} \pi(\boldsymbol{\beta}|\sigma^2, \boldsymbol{\gamma})\pi(\sigma^2|\boldsymbol{\gamma})\pi(\boldsymbol{\gamma}|\delta)\pi(\delta). \tag{8}$$

The EMVS approach is efficient in identifying potential sparse high posterior probability submodels on high-dimensional regression (Rořková and George, 2014) and classification problem (McDermott et al., 2016). These characteristics are helpful for large neural network computation, thus we refer the stochastic version of the EMVS algorithm as Expectation Stochastic-Maximization (ESM).

## 3.2 EXPECTATION STOCHASTIC-MAXIMIZATION

Due to the existence of latent variables, optimizing $\pi(\boldsymbol{\beta}, \sigma^2, \delta|\mathcal{B})$ directly is difficult. We instead iteratively optimize the "complete-data" posterior $\log \pi(\boldsymbol{\beta}, \sigma^2, \delta, \boldsymbol{\gamma}|\mathcal{B})$, where the latent indicator $\boldsymbol{\gamma}$ is treated as "missing data".

More precisely, the ESM algorithm is implemented by iteratively increasing the objective function

$$Q(\boldsymbol{\beta}, \sigma, \delta|\boldsymbol{\beta}^{(k)}, \sigma^{(k)}, \delta^{(k)}) = \mathbb{E}_{\boldsymbol{\gamma}|\cdot}\left[\log \pi(\boldsymbol{\beta}, \sigma, \delta, \boldsymbol{\gamma}|\mathcal{B})|\boldsymbol{\beta}^{(k)}, \sigma^{(k)}, \delta^{(k)}, \mathcal{B}\right], \tag{9}$$

where $\mathbb{E}_{\boldsymbol{\gamma}|\cdot}$ denotes the conditional expectation $\mathbb{E}_{\boldsymbol{\gamma}|\beta^{(k)},\sigma^{(k)},\delta^{(k)},\mathcal{B}}(\cdot)$. Given $(\boldsymbol{\beta}^{(k)},\sigma^{(k)},\delta^{(k)})$ at the k-th iteration, we first compute the expectation of $Q$, then alter $(\boldsymbol{\beta},\sigma,\delta)$ to optimize it.

For the conjugate spike-slab hierarchical prior formulation, the objective function $Q$ is of the form

$$Q(\boldsymbol{\beta},\sigma,\delta|\boldsymbol{\beta}^{(k)},\sigma^{(k)},\delta^{(k)}) = C + Q_1(\beta,\sigma|\boldsymbol{\beta}^{(k)},\sigma^{(k)},\delta^{(k)}) + Q_2(\delta|\boldsymbol{\beta}^{(k)},\sigma^{(k)},\delta^{(k)}), \qquad (10)$$

where

$$Q_1(\boldsymbol{\beta},\sigma|\boldsymbol{\beta}^{(k)},\sigma^{(k)},\delta^{(k)}) = \overbrace{\frac{N}{n}\log\pi(\mathcal{B}|\boldsymbol{\beta},\sigma^2)}^{\text{log likelihood}} - \frac{\overbrace{\sum_{j=1}^{p}\beta_j^2\mathbb{E}_{\boldsymbol{\gamma}|\cdot}\left[\frac{1}{v_0(1-\gamma_j)+v_1\gamma_j}\right]}^{\text{spike-and-slab priors in the "sparse" layer, e.g. FC1}}}{2\sigma^2}$$
$$- \underbrace{\frac{\sum_{j=p+1}^{p+u}\beta_j^2}{2}}_{\text{Gaussian priors in other layers}} - \frac{p+\nu+2}{2}\log(\sigma^2) - \frac{\nu\lambda}{2\sigma^2} \qquad (11)$$

and

$$Q_2(\delta|\boldsymbol{\beta}^{(k)},\sigma^{(k)},\delta^{(k)}) = \sum_{j=1}^{p}\log\left(\frac{\delta}{1-\delta}\right)\mathbb{E}_{\boldsymbol{\gamma}|\cdot}\gamma_j + (a-1)\log(\delta) + (p+b-1)\log(1-\delta). \qquad (12)$$

### 3.2.1 THE E-STEP

The physical meaning of $\mathbb{E}_{\boldsymbol{\gamma}|\cdot}\gamma_j$ in $Q_2$ is the probability $\rho_j$, where $\boldsymbol{\rho}\in\mathbb{R}^p$, of $\beta_j$ having a large effect on the model. Formally, we have

$$\rho_j = \mathbb{E}_{\boldsymbol{\gamma}|\cdot}\gamma_j = \mathrm{P}(\gamma_j=1|\boldsymbol{\beta}^{(k)},\sigma^{(k)},\delta^{(k)}) = \frac{a_j}{a_j+b_j}, \qquad (13)$$

where $a_j = \pi(\beta_j^{(k)}|\gamma_j=1)\mathrm{P}(\gamma_j=1|\delta^{(k)})$ and $b_j = \pi(\beta_j^{(k)}|\gamma_j=0)\mathrm{P}(\gamma_j=0|\delta^{(k)})$. The choice of Bernoulli prior enables us to use $\mathrm{P}(\gamma_j=1|\delta^{(k)}) = \delta^{(k)}$.

The other conditional expectation comes from a weighted average $\kappa_j$, where $\boldsymbol{\kappa}\in\mathbb{R}^p$.

$$\kappa_j = \mathbb{E}_{\boldsymbol{\gamma}|\cdot}\left[\frac{1}{v_0(1-\gamma_j)+v_1\gamma_j}\right] = \frac{\mathbb{E}_{\boldsymbol{\gamma}|\cdot}(1-\gamma_j)}{v_0} + \frac{\mathbb{E}_{\boldsymbol{\gamma}|\cdot}\gamma_j}{v_1} = \frac{1-\rho_j}{v_0} + \frac{\rho_j}{v_1}. \qquad (14)$$

### 3.2.2 THE STOCHASTIC M-STEP

Since there is no closed-form optimal solution for $\boldsymbol{\beta}$ here, to optimize $Q_1$ with respect to $\boldsymbol{\beta}$, we use Adam (Kingma and Ba, 2014), a popular algorithm with adaptive learning rates, to train the model.

In order to optimize $Q_1$ with respect to $\sigma$, by denoting $\mathrm{diag}\{\kappa_i\}_{i=1}^{p}$ as $\boldsymbol{\mathcal{V}}$, following the formulation in McDermott et al. (2016) and Ročková and George (2014) we have:

$$\sigma^{(k+1)} = \begin{cases} \sqrt{\dfrac{\frac{N}{n}\sum_{i\in\mathcal{S}}\left(y_i-\psi(\boldsymbol{x}_i;\boldsymbol{\beta}^{(k+1)})\right)^2 + ||\boldsymbol{\mathcal{V}}^{1/2}\boldsymbol{\beta}^{(k+1)}||^2 + \nu\lambda}{N+p+\nu}} & \text{(regression)}, \\[4ex] \sqrt{\dfrac{\left\|\left(\boldsymbol{\mathcal{V}}^{1/2}\boldsymbol{\beta}^{(k+1)}\right)\right\|^2 + \nu\lambda}{p+\nu+2}} & \text{(classification)}. \end{cases} \qquad (15)$$

To optimize $Q_2$, a closed-form solution can be derived from Eq.(12) and Eq.(13).

$$\delta^{(k+1)} = \arg\max_{\delta\in\mathbb{R}} Q_2(\delta|\boldsymbol{\beta}^{(k)},\sigma^{(k)},\delta^{(k)}) = \frac{\sum_{j=1}^{p}\rho_j + a - 1}{a+b+p-2}. \qquad (16)$$

---

**Algorithm 1** SGLD-SA with spike and slab priors

---

**Inputs:** $\{\epsilon^{(k)}\}_{k=1}^{\mathcal{T}}$, $\{\omega^{(k)}\}_{k=1}^{\mathcal{T}}$, $\tau$, a, b, $\nu$, $\lambda$, $v_0$, $v_1$
**Outputs:** $\{\boldsymbol{\beta}^{(k)}\}_{k=1}^{\mathcal{T}}$
**Initialize:** $\boldsymbol{\beta}^{(1)}$, $\boldsymbol{\rho}^{(1)}$, $\boldsymbol{\kappa}^{(1)}$, $\sigma^{(1)}$ and $\delta^{(1)}$ from scratch
**for** $k \leftarrow 1 : \mathcal{T}$ **do**
    Sample mini-batch $\mathcal{B}^{(k)}$
    **Sampling**
    $\boldsymbol{\beta}^{(k+1)} \leftarrow \boldsymbol{\beta}^{(k)} + \epsilon^{(k)} \nabla Q(\boldsymbol{\beta}, \sigma, \delta | \boldsymbol{\beta}^{(k)}, \sigma^{(k)}, \delta^{(k)}) + \mathcal{N}(0, 2\epsilon^{(k)}\tau^{-1})$
    **Optimization**
    Compute $\boldsymbol{\rho}^{(k+1)}$ in Eq.(13), perform SA: $\boldsymbol{\rho}^{(k+1)} \leftarrow (1 - \omega^{(k+1)})\boldsymbol{\rho}^{(k)} + \omega^{(k+1)}\boldsymbol{\rho}^{(k+1)}$
    Compute $\boldsymbol{\kappa}^{(k+1)}$ in Eq.(14), perform SA: $\boldsymbol{\kappa}^{(k+1)} \leftarrow (1 - \omega^{(k+1)})\boldsymbol{\kappa}^{(k)} + \omega^{(k+1)}\boldsymbol{\kappa}^{(k+1)}$
    Compute $\sigma^{(k+1)}$ in Eq.(15), perform SA: $\sigma^{(k+1)} \leftarrow (1 - \omega^{(k+1)})\sigma^{(k)} + \omega^{(k+1)}\sigma^{(k+1)}$
    Compute $\delta^{(k+1)}$ in Eq.(16), perform SA: $\delta^{(k+1)} \leftarrow (1 - \omega^{(k+1)})\delta^{(k)} + \omega^{(k+1)}\delta^{(k+1)}$
**end for**

---

### 3.3 SG-MCMC-SA

The EMVS algorithm is designed for linear regression models, although the idea can be extended to nonlinear models. However, when extending to nonlinear models, such as DNNs, the M-step will not have a closed-form update anymore. A trivial implementation of the M-step will likely cause a local-trap problem. To tackle this issue, we replace the E-step and the M-step by SG-MCMC with the prior hyperparameters tuned via stochastic approximation (Benveniste et al., 1990):

$$
\begin{aligned}
\boldsymbol{\beta}^{(k+1)} &= \boldsymbol{\beta}^{(k)} + \epsilon^{(k)} \nabla \tilde{L}(\boldsymbol{\beta}^{(k)}, \boldsymbol{\theta}^{(k)}) + \mathcal{N}(0, 2\epsilon^{(k)}\tau^{-1}), \\
\boldsymbol{\theta}^{(k+1)} &= \boldsymbol{\theta}^{(k)} + \omega^{(k+1)}\left(g_{\boldsymbol{\theta}^{(k)}}(\boldsymbol{\beta}^{(k+1)}) - \boldsymbol{\theta}^{(k)}\right),
\end{aligned}
\tag{17}
$$

where $g_{\boldsymbol{\theta}^{(k)}}(\boldsymbol{\beta}^{(k+1)}) - \boldsymbol{\theta}^{(k)}$ is the gradient-like function in stochastic approximation (see details in Appendix A.2), $g_{\boldsymbol{\theta}}(\cdot)$ is the mapping detailed in Eq.(13), Eq.(14), Eq.(15) and Eq.(16) to derive the optimal $\boldsymbol{\theta}$ based on the current $\boldsymbol{\beta}$, the step size $\omega^{(k)}$ can be set as $A(k + Z)^{-\alpha}$ with $\alpha \in (0.5, 1]$. The interpretation of this algorithm is that we sample $\boldsymbol{\beta}^{(k+1)}$ from $\tilde{L}(\boldsymbol{\beta}^{(k)}, \boldsymbol{\theta}^{(k)})$ and adaptively optimize $\boldsymbol{\theta}^{(k+1)}$ from the mapping $g_{\boldsymbol{\theta}^{(k)}}$. We expect to obtain an augmented sequence as follows:

$$
\boldsymbol{\beta}^{(1)}, \boldsymbol{\theta}^{(1)}; \boldsymbol{\beta}^{(2)}, \boldsymbol{\theta}^{(2)}; \boldsymbol{\beta}^{(3)}, \boldsymbol{\theta}^{(3)}; \ldots
\tag{18}
$$

We show the (local) $L^2$ convergence rate of SGLD-SA below and present the details in Appendix B.

**Theorem 1** ($L^2$ convergence rate). *For any $\alpha \in (0, 1]$ and any compact subset $\boldsymbol{\Omega} \in \mathbb{R}^{2p+2}$, under assumptions in Appendix B.1, the algorithm satisfies: there exists a constant $\lambda$ such that*

$$
\mathbb{E}\left[\|\boldsymbol{\theta}^{(k)} - \boldsymbol{\theta}^*\|^2 \mathbf{I}(k \leq t(\boldsymbol{\Omega}))\right] \leq \lambda \omega^{(k)},
$$

*where $t(\boldsymbol{\Omega}) = \inf\{k : \boldsymbol{\theta}^{(k)} \notin \boldsymbol{\Omega}\}$ and $\omega^{(k)} = A(k + Z)^{-\alpha} \sim \mathcal{O}(k^{-\alpha})$.*

**Corollary 1.** *For any $\alpha \in (0, 1]$ and any compact subset $\boldsymbol{\Omega} \in \mathbb{R}^{2p+2}$, under assumptions in Appendix B.2, the distribution of $\boldsymbol{\beta}^{(k)}$ in (18) converges weakly to the invariant distribution $e^{L(\boldsymbol{\beta}, \boldsymbol{\theta}^*)}$ as $\epsilon \to 0$.*

The key to guaranteeing the consistency of the latent variable estimators is from stochastic approximation and the fact of $\sum_{i=1}^{\infty} \omega^{(i)} = \infty$. The non-convex optimization of SGLD (Raginsky et al., 2017; Xu et al., 2018), in particular the ability to escape shallow local optima (Zhang et al., 2017), ensures the robust optimization of the latent variables. Furthermore, the mappings of Eq.(13), Eq.(14), Eq.(15) and Eq.(16) all satisfy the assumptions on $g$ in a compact subset $\boldsymbol{\Omega}$ (Appendix B), which enable us to apply theorem 1 to SGLD-SA. Because SGHMC proposed by Chen et al. (2014) is essentially a second-order Langevin dynamics and yields a stationary distribution given an accurate estimation of the error variance from the stochastic gradient, the property of SGLD-SA also applies to SGHMC-SA.

**Corollary 2.** *For any $\alpha \in (0, 1]$ and any compact subset $\boldsymbol{\Omega} \in \mathbb{R}^{2p+2}$, under assumptions in Appendix B.2, the distribution of $\boldsymbol{\beta}^{(k)}$ from SG-MCMC-SA converges weakly to the invariant distribution $\pi(\boldsymbol{\beta}, \boldsymbol{\rho}^*, \boldsymbol{\kappa}^*, \sigma^*, \delta^* | \mathcal{D})$ as $\epsilon \to 0$.*

Table 1: Predictive errors in linear regression based on a test set considering different $v_0$ and $\sigma$

| MAE / MSE | $v_0=10^{-3}, \sigma=0.4$ | $v_0=10^{-3}, \sigma=0.5$ | $v_0=10^{-2}, \sigma=0.4$ | $v_0=10^{-2}, \sigma=0.5$ |
|---|---|---|---|---|
| SGLD-SA | **1.32 / 2.85** | **1.34 / 2.90** | **1.34 / 2.92** | 1.37 / **2.93** |
| EMVS | 1.43 / 3.19 | 3.04 / 13.6 | 3.40 / 18.8 | **1.33** / 2.95 |
| SGLD | 4.98 / 42.6 | 4.98 / 42.6 | 4.98 / 42.6 | 4.98 / 42.6 |

### 3.4 POSTERIOR APPROXIMATION

The posterior average given decreasing learning rates can be approximated through the weighted sample average $\mathbb{E}[\psi(\boldsymbol{\beta})] = \frac{\sum_{k=1}^{\mathcal{T}} \epsilon^{(k)} \psi(\boldsymbol{\beta}^{(k)})}{\sum_{k=1}^{\mathcal{T}} \epsilon^{(k)}}$ (Welling and Teh, 2011) to avoid over-emphasizing the tail end of the sequence and reduce the variance of the estimator. Teh et al. (2015); Chen et al. (2015) showed a theoretical optimal learning rate $\epsilon^{(k)} \propto k^{-1/3}$ for SGLD and $k^{-1/5}$ for SGHMC to achieve faster convergence for posterior average, which are used in Sec. 4.1 and Sec. 4.2 respectively.

## 4 EXPERIMENTS

### 4.1 SIMULATION OF LARGE-P-SMALL-N REGRESSION

SGLD-SA can be applied to the (logistic) linear regression cases, as long as $u = 0$ in Eq.(11). We conduct the linear regression experiments with a dataset containing $n = 100$ observations and $p = 1000$ predictors. $\mathcal{N}_p(0, \boldsymbol{\Sigma})$ is chosen to simulate the predictor values $\boldsymbol{X}$ (training set) where $\boldsymbol{\Sigma} = (\Sigma)_{i,j=1}^p$ with $\Sigma_{i,j} = 0.6^{|i-j|}$. Response values $\boldsymbol{y}$ are generated from $\boldsymbol{X}\boldsymbol{\beta} + \boldsymbol{\eta}$, where $\boldsymbol{\beta} = (\beta_1, \beta_2, \beta_3, 0, 0, ..., 0)'$ and $\boldsymbol{\eta} \sim \mathcal{N}_n(\boldsymbol{0}, 3\boldsymbol{I}_n)$. To make the simulation in Ročková and George (2014) more challenging, we assume $\beta_1 \sim \mathcal{N}(3, \sigma_c^2)$, $\beta_2 \sim \mathcal{N}(2, \sigma_c^2)$, $\beta_3 \sim \mathcal{N}(1, \sigma_c^2)$, $\sigma_c = 0.2$.

We introduce some hyperparameters, but most of them are uninformative, e.g. $\nu \in \{10^{-3}, 1, 10^3\}$ makes little difference in the test set performance. Sensitivity analysis shows that three hyperparameters are important: $v_0$, $a$ and $\sigma$, which are used to identify and regularize the over-fitted space. We fix $\tau = 1, \lambda = 1, \nu = 1, v_1 = 100, \delta = 0.5, b = p$ and set $a = 1$. The learning rates for SGLD-SA and SGLD are set to $\epsilon^{(k)} = 0.001 \times k^{-\frac{1}{3}}$, and the step size $\omega^{(k)} = 0.1 \times (k + 1000)^{-\frac{3}{4}}$. We vary $v_0$ and $\sigma$ to show the robustness of SGLD-SA with respect to different initializations.

To implement SGLD-SA, we perform stochastic gradient optimization by randomly selecting 50 observations and calculating the corresponding gradient in each iteration. We simulate $500,000$ samples from the posterior distribution and at the same time keep optimizing the latent variables. EMVS is implemented with $\boldsymbol{\beta}$ directly optimized each time. We also simulate a group of the test set with 1000 observations (display 50 in Fig. 2(e)) in the same way as generating the training set to evaluate the generalizability of our algorithm. Tab.1 shows that **EMVS frequently fails given bad initializations**, while **SGLD-SA** is fairly **robust to the hyperparameters**. In addition, from Fig. 2(a), Fig. 2(b) and Fig.2(c), we can see **SGLD-SA is the only algorithm among the three that quantifies the uncertainties of $\beta_1$, $\beta_2$ and $\beta_3$ reasonably** and always gives more accurate posterior average (Fig.2(f)); by contrast, the estimated response values $\boldsymbol{y}$ from SGLD is close to the true values in the training set (Fig.2(d)), but are far away from them in the testing set (2(e)), **indicating the over-fitting problem of SGLD without proper regularization**.

For the simulation of **SGLD-SA** in logistic regression to demonstrate the advantage of SGLD-SA over SGLD and ESM, we leave the results in Appendix C.

### 4.2 CLASSIFICATION PROBLEM

We implement all the algorithms in Pytorch (Paszke et al., 2017) and run the experiments on GeForce GTX 1080 GPUs. The first DNN we use is a standard 2-Conv-2-FC CNN model (Fig.1) of 670K parameters (see details in Appendix D.1). The first set of experiments is to compare methods on the same model without using other complication, such as data augmentation (DA) or batch normalization (BN) (Ioffe and Szegedy, 2015). We refer to the general CNN without dropout as **Vanilla**, with 50% dropout rate applied to the green neurons (Fig.1) as **Dropout**. Vanilla and Dropout models are trained

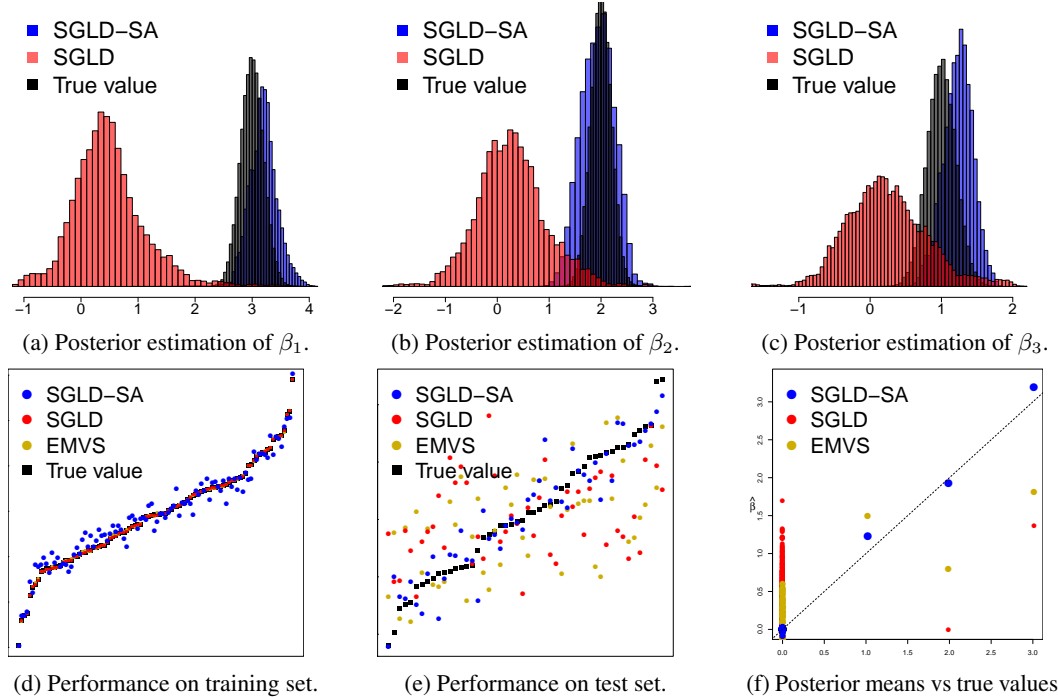

Figure 2: Linear regression simulation when $v_0 = 10^{-3}$ and $\sigma = 0.5$.

with Adam (Kingma and Ba, 2014) with Pytorch default parameters (with learning rate 0.001). We use SGHMC as a benchmark method as it is also sampling-based and has a close relationship with the popular momentum based optimization approaches in DNN. **SGHMC-SA** differs from **SGHMC** in that **SGHMC-SA** applies the spike-and-slab priors to the **FC1** layer while **SGHMC** just uses the standard normal priors. The hyperparameters $v_1 = 1, v_0 = 1 \times 10^{-3}$ and $\sigma = 0.1$ in **SGHMC-SA** are used to regularize the over-fitted space, and $a, b$ are set to $p$ to obtain a moderate "sparsity" to resemble dropout, the step size is $\omega^{(k)} = 0.1 \times (k + 1000)^{-\frac{3}{4}}$. We use training batch size 1000 and a thinning factor 500 to avoid a cumbersome system, and the posterior average is applied to each Bayesian model. Temperatures are tuned to achieve better results (Appendix D.2).

The four CNN models are tested on the classical MNIST and the newly introduced Fashion MNIST (FMNIST) (Xiao et al., 2017) dataset. Performance of these models is shown in Tab.2. Compared with SGHMC, our SGHMC-SA outperforms SGHMC on both datasets. We notice the posterior averages from SGHMC-SA and SGHMC obtain much better performance than Vanilla and Dropout. **Without using either DA or BN, SGHMC-SA achieves 99.60% which even outperforms some state-of-the-art models**, such as stochastic pooling (99.53%) (Zeiler and Fergus, 2013), Maxout Network (99.55%) (Goodfellow et al., 2013) and pSGLD (99.55%) (Li et al., 2016) . In **F-MNIST**, **SGHMC-SA** obtains 93.01% accuracy, outperforming all other competing models.

To further test the maximum performance of SGHMC-SA, we apply DA and BN to the following experiments (see details in Appendix D.3) and refer the datasets with DA as $a$MNIST and $a$FMNIST. All the experiments are conducted using a 2-Conv-BN-3-FC CNN of 490K parameters. Using this model, we obtain **99.75%** on $a$MNIST (300 epochs) and 94.38% on $a$FMNIST (1000 epochs). The results are noticeable because posterior model averaging is essentially conducted on a single Bayesian neural network. We also conduct the experiments based on the ensemble of five networks and refer them as $a$MNIST-5 and $a$FMNIST-5 in Tab. 2. We achieve **99.79% on $a$MNIST-5** using 5 small Bayesian neural networks each with 2 thinning samples (4 thinning samples in $a$FMNIST-5), which is comparable with the **state-of-the-art performance** (Wan et al., 2013).

### 4.3 DEFENSES AGAINST ADVERSARIAL ATTACKS

Continuing with the setup in Sec. 4.2, the third set of experiments focus on evaluating model robustness. We expect less robust models perform considerably well on a certain dataset due to

Table 2: Classification accuracy on MNIST and Fashion MNIST using small networks

| DATASET | MNIST | $a$MNIST | $a$MNIST-5 | FMNIST | $a$FMNIST | $a$FMNIST-5 |
|---|---|---|---|---|---|---|
| VANILLA | 99.31 | 99.54 | 99.75 | 92.73 | 93.14 | 94.48 |
| DROPOUT | 99.38 | 99.56 | 99.74 | 92.81 | 93.35 | 94.53 |
| SGHMC | 99.55 | 99.71 | 99.77 | 92.93 | 94.29 | 94.64 |
| **SGHMC-SA** | **99.60** | **99.75** | **99.79** | **93.01** | **94.38** | **94.78** |

(a) $\zeta = ..., 0.3, ....$     (b) MNIST     (c) $\zeta = ..., 0.3, ....$     (d) F-MNIST

Figure 3: Adversarial test accuracies based on adversarial images of different levels

over-tuning; however, as the degree of adversarial attacks increases, the performance decreases sharply. In contrast, more robust models should be less affected by these adversarial attacks.

We apply the *Fast Gradient Sign* method (Goodfellow et al., 2014b) to generate the adversarial examples with one single gradient step as in Papernot et al. (2016)'s study:

$$\boldsymbol{x}_{adv} \leftarrow \boldsymbol{x} - \zeta \cdot \text{sign}\{\delta_{\boldsymbol{x}} \max_{\text{y}} \log \text{P}(\text{y} \,|\, \boldsymbol{x})\},$$

where $\zeta$ ranges from $0.1, 0.2, \ldots, 0.5$ to control the different levels of adversarial attacks.

Similar to the setup in the adversarial experiments by Li and Gal (2017), we normalize the adversarial images by clipping to the range $[0, 1]$. As shown in Fig. 3(b) and Fig.3(d), there is no significant difference among all the four models in the early phase. As the degree of adversarial attacks arises, the images become vaguer as shown in Fig.3(a) and Fig.3(c). In this scenario the performance of **Vanilla** decreases rapidly, reflecting its poor defense against adversarial attacks, while **Dropout** performs better than **Vanilla**. But **Dropout** is still significantly worse than the sampling based methods **SGHMC-SA** and **SGHMC**. The advantage of **SGHMC-SA** over **SGHMC** becomes more significant when $\zeta > 0.25$.

In the case of $\zeta = 0.5$ in MNIST where the images are hardly recognizable, both **Vanilla** and **Dropout** models fail to identify the right images and their predictions are as worse as random guesses. However, **SGHMC-SA** model achieves roughly **11%** higher than these two models and **1%** higher than **SGHMC**, which demonstrates the strong robustness of our proposed **SGHMC-SA**. Overall, **SGHMC-SA** always yields the most robust performance.

## 5   CONCLUSION AND FUTURE RESEARCH

We propose a mixed sampling-optimization method called **SG-MCMC-SA** to efficiently sample from complex DNN posteriors with latent variables and prove its convergence. By adaptively searching and penalizing the over-fitted parameter space, the proposed method improves the generalizability of deep neural networks. This method is less affected by the hyperparameters, achieves higher prediction accuracy over the traditional SG-MCMC methods in both simulated examples and real applications and shows more robustness towards adversarial attacks.

Interesting future directions include applying SG-MCMC-SA towards popular large deep learning models such as the residual network (He et al., 2015) on CIFAR-10 and CIFAR-100, combining active learning and uncertainty quantification to learn from datasets of smaller size and proving posterior consistency and the consistency of variable selection under various shrinkage priors concretely.

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

# A REVIEW

## A.1 FOKKER-PLANCK EQUATION

The Fokker-Planck equation (FPE) can be formulated from the time evolution of the conditional distribution for a stationary random process. Denoting the probability density function of the random process at time t by $q(t, \beta)$, where $\beta$ is the parameter, the stochastic dynamics is given by

$$\partial_t q(t, \beta) = \partial_\beta(\partial_\beta(-L(\beta))q(t, \beta)) + \partial_\beta^2 q(t, \beta). \tag{19}$$

Let $q(\beta) = \lim_{t \to \infty} q(t, \beta)$. If $\lim_{t \to \infty} \partial_t q(t, \beta) = 0$, then

$$\begin{aligned}
&\lim_{t \to \infty} \partial_\beta(\partial_\beta(-L(\beta))q(t, \beta)) + \partial_\beta^2 q(t, \beta) = 0 \\
&\Rightarrow \partial_\beta \left( \partial_\beta \left( -L(\beta)q(\beta) \right) \right) + \partial_\beta^2 q(\beta) = 0 \\
&\Leftrightarrow \partial_\beta \left( \partial_\beta(-L(\beta))q(\beta) + \partial_\beta q(\beta) \right) = 0 \\
&\Leftrightarrow \partial_\beta \left( q(\beta) \left( \partial_\beta(-L(\beta) + \partial_\beta \log q(\beta)) \right) \right) = 0.
\end{aligned} \tag{20}$$

Therefore, $\partial_\beta(-L(\beta)) + \partial_\beta \log(q(\beta)) = 0$, which implies $q(\beta) \propto e^{L(\beta)}$. In other words, $\lim_{t \to \infty} q(t, \beta) \propto e^{L(\beta)}$, i.e. $q(t, \beta)$ gradually converges to the Bayesian posterior $e^{L(\beta)}$.

## A.2 STOCHASTIC APPROXIMATION

### A.2.1 SPECIAL CASE: ROBBINS–MONRO ALGORITHM

Robbins–Monro algorithm is the first stochastic approximation algorithm to deal with the root finding problem. Given the random output of $H(\boldsymbol{\theta}, \boldsymbol{\beta})$ with respect to $\boldsymbol{\beta}$, our goal is to find $\boldsymbol{\theta}^*$ such that

$$h(\boldsymbol{\theta}^*) = \mathbb{E}_{\boldsymbol{\theta}^*}[H(\boldsymbol{\theta}^*, \boldsymbol{\beta})] = 0, \tag{21}$$

where $\mathbb{E}_{\boldsymbol{\theta}^*}$ denotes the expectation with respect to the distribution of $\boldsymbol{\beta}$ given parameter $\boldsymbol{\theta}^*$. To implement the Robbins–Monro Algorithm, we can generate iterates of the form as follows :

$$\boldsymbol{\theta}^{(k+1)} = \boldsymbol{\theta}^{(k)} + \omega^{(k+1)} H(\boldsymbol{\theta}^{(k)}, \boldsymbol{\beta}^{(k+1)}).$$

Note that in this algorithm, $H(\boldsymbol{\theta}, \boldsymbol{\beta})$ is the **unbiased estimator** of $h(\boldsymbol{\theta})$, that is

$$\mathbb{E}_{\boldsymbol{\theta}^{(k)}}[H(\boldsymbol{\theta}^{(k)}, \boldsymbol{\beta}^{(k+1)}) - h(\boldsymbol{\theta}^{(k)})|\mathcal{F}_k] = 0. \tag{22}$$

If there exists an antiderivative $Q(\boldsymbol{\theta}, \boldsymbol{\beta})$ that satisfies $H(\boldsymbol{\theta}, \boldsymbol{\beta}) = \frac{\partial Q(\boldsymbol{\theta}, \boldsymbol{\beta})}{\partial \boldsymbol{\theta}}$ and $E_{\boldsymbol{\theta}}[Q(\boldsymbol{\theta}, \boldsymbol{\beta})]$ is concave, it is equivalent to solving the stochastic optimization problem $\max_{\boldsymbol{\theta} \in \Theta} E_{\boldsymbol{\theta}}[Q(\boldsymbol{\theta}, \boldsymbol{\beta})]$.

### A.2.2 GENERAL STOCHASTIC APPROXIMATION

In contrast to Eq.(21) and Eq.(22), the general stochastic approximation algorithm is intent on solving the root of the integration equation

$$\begin{aligned}
h(\boldsymbol{\theta}) &= \lim_{k \to \infty} \mathbb{E}_{\boldsymbol{\theta}}[H(\boldsymbol{\theta}, \boldsymbol{\beta}^{(k+1)})|\mathcal{F}_k] \\
&= \lim_{k \to \infty} \int H(\boldsymbol{\theta}, \boldsymbol{\beta})\Pi_{\boldsymbol{\theta}}(\boldsymbol{\beta}^{(k)}, d\boldsymbol{\beta}) \\
&= \int H(\boldsymbol{\theta}, \boldsymbol{\beta})f_{\boldsymbol{\theta}}(d\boldsymbol{\beta}) = 0,
\end{aligned} \tag{23}$$

where $\boldsymbol{\theta} \in \Theta$, $\boldsymbol{\beta} \in B$, for a subset $\mathbf{B} \in B$, the transition kernel $\Pi_{\boldsymbol{\theta}}(\boldsymbol{\beta}^{(k)}, \mathbf{B})$, which converges to the invariant distribution $f_{\boldsymbol{\theta}}(\boldsymbol{\beta})$, satisfies that $P[\boldsymbol{\beta}^{(k+1)} \in \mathbf{B}|\mathcal{F}_k] = \Pi_{\boldsymbol{\theta}}(\boldsymbol{\beta}^{(k)}, \mathbf{B})$. The stochastic approximation algorithm is an iterative recursive algorithm consisting of two steps:

(1) Generate $\boldsymbol{\beta}^{(k+1)}$ from the transition kernel $\Pi_{\boldsymbol{\theta}^{(k)}}(\boldsymbol{\beta}^{(k)}, \cdot)$, which admits $f_{\boldsymbol{\theta}^{(k)}}(\boldsymbol{\beta})$ as the invariant distribution,

(2) Update $\boldsymbol{\theta}^{(k+1)} = \boldsymbol{\theta}^{(k)} + \omega^{(k+1)} H(\boldsymbol{\theta}^{(k)}, \boldsymbol{\beta}^{(k+1)})$.

Then we have $\mathbb{E}_{\boldsymbol{\theta}^{(k)}}[H(\boldsymbol{\theta}^{(k)}, \boldsymbol{\beta}^{(k+1)})|\mathcal{F}_k] = \int H(\boldsymbol{\theta}^{(k)}, \boldsymbol{\beta})\Pi_{\boldsymbol{\theta}}(\boldsymbol{\beta}^{(k)}, d\boldsymbol{\beta})$. Compared with Eq.(22), $h(\boldsymbol{\theta}^{(k)}) - \mathbb{E}_{\boldsymbol{\theta}^{(k)}}[H(\boldsymbol{\theta}^{(k)}, \boldsymbol{\beta}^{(k+1)})|\mathcal{F}_k]$ is not equal to 0 but decays to 0 as $k \to \infty$.

To summarise, $H(\boldsymbol{\theta}, \boldsymbol{\beta})$ **is a biased estimator of** $h(\boldsymbol{\theta})$ **in finite steps, but as** $k \to \infty$**, the bias decreases to 0**. In the SG-MCMC-SA algorithm (17), $\mathbf{H}(\boldsymbol{\theta}^{(\mathbf{k})}, \boldsymbol{\beta}^{(\mathbf{k+1})})$ is defined by $\mathbf{g}_{\boldsymbol{\theta}^{(\mathbf{k})}}(\boldsymbol{\beta}^{(\mathbf{k+1})}) - \boldsymbol{\theta}^{(\mathbf{k})}$.

# B    CONVERGENCE ANALYSIS

## B.1    CONVERGENCE OF HIDDEN VARIABLES

The stochastic gradient Langevin Dynamics with a stochastic approximation adaptation (SGLD-SA) is a mixed half-optimization-half-sampling algorithm to handle complex Bayesian posterior with latent variables, e.g. the conjugate spike-slab hierarchical prior formulation. Each iteration of the algorithm consists of the following steps:

(1) Sample $\boldsymbol{\beta}^{(k+1)}$ using SGLD based on the current $\boldsymbol{\theta}^{(k)}$, i.e.

$$\boldsymbol{\beta}^{(k+1)} = \boldsymbol{\beta}^{(k)} + \epsilon\frac{\partial}{\partial\boldsymbol{\beta}}\tilde{L}(\boldsymbol{\beta}^{(k)}, \boldsymbol{\theta}^{(k)}) + \mathcal{N}(0, 2\epsilon\tau^{-1}); \tag{24}$$

(2) Optimize $\boldsymbol{\theta}^{(k+1)}$ from the following recursion

$$\begin{aligned}\boldsymbol{\theta}^{(k+1)} &= \boldsymbol{\theta}^{(k)} + \omega^{(k+1)}\left(g_{\boldsymbol{\theta}^{(k)}}(\boldsymbol{\beta}^{(k+1)}) - \boldsymbol{\theta}^{(k)}\right) \\ &= (1 - \omega^{(k+1)})\boldsymbol{\theta}^{(k)} + \omega^{(k+1)}\boldsymbol{g}_{\boldsymbol{\theta}^{(k)}}(\boldsymbol{\beta}^{(k+1)}),\end{aligned} \tag{25}$$

where $\boldsymbol{g}_{\boldsymbol{\theta}^{(k)}}(\cdot)$ is some mapping to derive the optimal $\boldsymbol{\theta}$ based on the current $\boldsymbol{\beta}$.

**Remark**: Define $\mathbf{H}(\boldsymbol{\theta}^{(\mathbf{k})}, \boldsymbol{\beta}^{(\mathbf{k+1})}) = \mathbf{g}_{\boldsymbol{\theta}^{(\mathbf{k})}}(\boldsymbol{\beta}^{(\mathbf{k+1})}) - \boldsymbol{\theta}^{(\mathbf{k})}$. In this formulation, our target is to find $\boldsymbol{\theta}^*$ such that $h(\boldsymbol{\theta}^*) = 0$, where $h(\boldsymbol{\theta}) := \int H(\boldsymbol{\theta}, \boldsymbol{\beta})f_{\boldsymbol{\theta}}(d\boldsymbol{\beta})$. $\mathbb{E}_{\boldsymbol{\theta}^{(k)}}[H(\boldsymbol{\theta}^{(k)}, \boldsymbol{\beta}^{(k+1)})|\mathcal{F}_k]$ converges to $h(\boldsymbol{\theta}^{(k)})$ as $k \to \infty$, this algorithm falls to the category of the general stochastic approximation.

GENERAL ASSUMPTIONS

To provide the local $L^2$ upper bound for SGLD-SA, we first lay out the following assumptions:

**Assumption 1** (Step size and Convexity). *$\{\omega^{(k)}\}_{k\in\mathrm{N}}$ is a positive decreasing sequence of real numbers such that*

$$\omega^{(k)} \to 0, \ \sum_{k=1}^{\infty}\omega^{(k)} = +\infty. \tag{26}$$

*There exist constant $\delta > 0$ and $\boldsymbol{\theta}^*$ such that for all $\boldsymbol{\theta} \in \boldsymbol{\Theta}$*

$$\langle\boldsymbol{\theta} - \boldsymbol{\theta}^*, h(\boldsymbol{\theta})\rangle \leq -\delta\|\boldsymbol{\theta} - \boldsymbol{\theta}^*\|^2, \tag{27}$$

*with additionally*

$$\lim_{k\to\infty}\inf 2\delta\frac{\omega^{(k)}}{\omega^{(k+1)}} + \frac{\omega^{(k+1)} - \omega^{(k)}}{\omega^{(k+1)2}} > 0. \tag{28}$$

*Then for any $\alpha \in (0, 1]$ and suitable A and B, a practical $\omega^{(k)}$ can be set as*

$$\omega^{(k)} = A(k + B)^{-\alpha} \tag{29}$$

**Assumption 2** (Existence of Markov transition kernel). *For any $\boldsymbol{\theta} \in \boldsymbol{\Theta}$, in the mini-batch sampling, there exists a family of noisy Kolmogorov operators $\{\Pi_{\boldsymbol{\theta}}\}$ approximating the operator (infinitesimal) of the Ito diffusion, such that every Kolmogorov operator $\Pi_{\boldsymbol{\theta}}$ corresponds to a single stationary distribution $f_{\boldsymbol{\theta}}$, and for any Borel subset $\boldsymbol{A}$ of $\boldsymbol{\beta}$, we have*

$$\mathrm{P}[\boldsymbol{\beta}^{(k+1)} \in \boldsymbol{A}|\mathcal{F}_k] = \Pi_{\boldsymbol{\theta}^{(k)}}(\boldsymbol{\beta}^{(k)}, \boldsymbol{A}).$$

**Assumption 3** (Compactness). *For any compact subset $\Omega$ of $\Theta$, we only consider $\theta \in \Omega$ such that*

$$\|\theta\| \leq C_0(\Omega) \tag{30}$$

*Note that the compactness assumption of the latent variable $\theta$ is not essential, the assumption that the variable is in the compact domain is not only reasonable, but also simplifies our proof.*

*In addition, there exists constants $C_1(\Omega)$ and $C_2(\Omega)$ so that*

$$\begin{aligned}
\mathbb{E}_{\theta^{(k)}}[\|H(\theta^{(k)}, \beta^{(k+1)})\|^2 | \mathcal{F}_k] &\leq C_1(\Omega)(1 + \|\theta^{(k)}\|^2) \\
&\leq C_2(\Omega)(1 + \|\theta^{(k)} - \theta^* + \theta^*\|^2) \\
&\leq C_2(\Omega)(1 + \|\theta^{(k)} - \theta^*\|^2)
\end{aligned} \tag{31}$$

**Assumption 4** (Solution of Poisson equation). *For all $\theta \in \Theta$, there exists a function $\mu_\theta$ on $\beta$ that solves the Poisson equation $\mu_\theta(\beta) - \Pi_\theta \mu_\theta(\beta) = H(\theta, \beta) - h(\theta)$, which follows that*

$$H(\theta^{(k)}, \beta^{(k+1)}) = h(\theta^{(k)}) + \mu_{\theta^{(k)}}(\beta^{(k+1)}) - \Pi_{\theta^{(k)}} \mu_{\theta^{(k)}}(\beta^{(k+1)}). \tag{32}$$

*For any compact subset $\Omega$ of $\Theta$, there exist constants $C_3(\beta, \Omega)$ and $C_4(\beta, \Omega)$ such that for all $\theta, \theta' \in \Omega$,*

$$\begin{aligned}
&(1) \; \|\Pi_\theta \mu_\theta\| \leq C_3(\Omega), \\
&(2) \; \|\Pi_\theta \mu_\theta - \Pi_{\theta'} \mu_{\theta'}\| \leq C_4(\Omega)\|\theta - \theta'\|,
\end{aligned} \tag{33}$$

**Remark**: For notation simplicity, we write $C_1(\Omega)$ as $C_1$, $C_2(\Omega)$ as $C_2$, ... in the following context.

Lemma 1 is a restatement of Lemma 25 (page 447) from Benveniste et al. (1990).

**Lemma 1.** *Suppose $k_0$ is an integer which satisfies with*

$$\inf_{k \geq k_0} \frac{\omega^{(k+1)} - \omega^{(k)}}{\omega^{(k)} \omega^{(k+1)}} + 2\delta - \omega^{(k+1)} C_2 > 0.$$

*Then for any $k > k_0$, the sequence $\{\Lambda_k^K\}_{k=k_0,\ldots,K}$ defined below is increasing and bounded by $2\omega^{(K)}$*

$$\Lambda_k^K = \begin{cases} 2\omega^{(k)} \prod_{j=k}^{K-1}(1 - 2\omega_{j+1}\delta + \omega_{j+1}^2 C_2) & \text{if } k < K, \\ 2\omega^{(K)} & \text{if } k = K. \end{cases} \tag{34}$$

Lemma 2 is an extension of Lemma 23 (page 245) from Benveniste et al. (1990)

**Lemma 2.** *There exist $\lambda_0$ and $k_0$ such that for all $\lambda \geq \lambda_0$ and $k \geq k_0$, the sequence $u^{(k)} = \lambda \omega^{(k)}$ satisfies*

$$u^{(k+1)} \geq (1 - 2\omega^{(k+1)}\delta + \omega^{{(k+1)}^2} C_2)u^{(k)} + \omega^{{(k+1)}^2} C_2 + \omega^{(k+1)} \overline{C}_1. \tag{35}$$

*Proof.* Replace $u^{(k)} = \lambda \omega^{(k)}$ in (35), we have

$$\lambda \omega^{(k+1)} \geq (1 - 2\omega^{(k+1)}\delta + \omega^{{(k+1)}^2} C_2)\lambda \omega^{(k)} + \omega^{{(k+1)}^2} C_2 + \omega^{(k+1)} \overline{C}_1. \tag{36}$$

From (28), we can denote a positive constant $\Delta_+$ as $\lim_{k \to \infty} \inf 2\delta \omega^{(k+1)} \omega^{(k)} + \omega^{(k+1)} - \omega^{(k)}$. Then (36) can be simplified as

$$\lambda(\Delta_+ - \omega^{{(k+1)}^2} \omega^{(k)} C_2) \geq \omega^{{(k+1)}^2} C_2 + \omega^{(k+1)} \overline{C}_1. \tag{37}$$

There exist $\lambda_0$ and $k_0$ such that for all $\lambda > \lambda_0$ and $k > k_0$, (37) holds. Note that in practical case when $\overline{C}_1$ is small, finding a suitable $\lambda_0$ will not be a problem. $\qquad \square$

**Theorem 1** ($L^2$ convergence rate). Suppose that Assumptions 1-4 hold, for any compact subset $\mathbf{\Omega} \in \mathbf{\Theta}$, the algorithm satisfies: there exists a constant $\lambda$ such that

$$\mathbb{E}\left[\|\boldsymbol{\theta}^{(k)} - \boldsymbol{\theta}^*\|^2 \mathrm{I}(k \leq t(\mathbf{\Omega}))\right] \leq \lambda \omega^{(k)},$$

where $t(\mathbf{\Omega}) = \inf\{k : \boldsymbol{\theta}^{(k)} \notin \mathbf{\Omega}\}$.

*Proof.* Denote $\boldsymbol{T}^{(k)} = \boldsymbol{\theta}^{(k)} - \boldsymbol{\theta}^*$, with the help of (25) and Poisson equation (32), we deduce that

$$
\begin{aligned}
\|\boldsymbol{T}^{(k+1)}\|^2 &= \|\boldsymbol{T}^{(k)}\|^2 + \omega^{(k+1)^2}\|H(\boldsymbol{\theta}^{(k)}, \boldsymbol{\beta}^{(k+1)})\|^2 + 2\omega^{(k+1)}\langle \boldsymbol{T}^{(k)}, H(\boldsymbol{\theta}^{(k)}, \boldsymbol{\beta}^{(k+1)})\rangle \\
&= \|\boldsymbol{T}^{(k)}\|^2 + \omega^{(k+1)^2}\|H(\boldsymbol{\theta}^{(k)}, \boldsymbol{\beta}^{(k+1)})\|^2 \\
&\quad + 2\omega^{(k+1)}\langle \boldsymbol{T}^{(k)}, h(\boldsymbol{\theta}^{(k)})\rangle + 2\omega^{(k+1)}\langle \boldsymbol{T}^{(k)}, \mu_{\boldsymbol{\theta}^{(k)}}(\boldsymbol{\beta}^{(k+1)}) - \Pi_{\boldsymbol{\theta}^{(k)}}\mu_{\boldsymbol{\theta}^{(k)}}(\boldsymbol{\beta}^{(k+1)})\rangle \\
&= \|\boldsymbol{T}^{(k)}\|^2 + \mathrm{D1} + \mathrm{D2} + \mathrm{D3}.
\end{aligned}
$$

First of all, according to (31) and (27), we have

$$\omega^{(k+1)^2}\|H(\boldsymbol{\theta}^{(k)}, \boldsymbol{\beta}^{(k+1)})\|^2 \leq \omega^{(k+1)^2}C_2(1 + \|\boldsymbol{T}^{(k)}\|^2), \tag{D1}$$

$$2\omega^{(k+1)}\langle \boldsymbol{T}^{(k)}, h(\boldsymbol{\theta}^{(k)})\rangle \leq -2\omega^{(k+1)}\delta\|\boldsymbol{T}^{(k)}\|^2, \tag{D2}$$

Conduct the decomposition of D3 similar to Theorem 24 (p.g. 246) from Benveniste et al. (1990) and Lemma A.5 (Liang, 2010).

$$
\begin{aligned}
&\mu_{\boldsymbol{\theta}^{(k)}}(\boldsymbol{\beta}^{(k+1)}) - \Pi_{\boldsymbol{\theta}^{(k)}}\mu_{\boldsymbol{\theta}^{(k)}}(\boldsymbol{\beta}^{(k+1)}) \\
&= \left(\mu_{\boldsymbol{\theta}^{(k)}}(\boldsymbol{\beta}^{(k+1)}) - \Pi_{\boldsymbol{\theta}^{(k)}}\mu_{\boldsymbol{\theta}^{(k)}}(\boldsymbol{\theta}^{(k)})\right) \\
&\quad + \left(\Pi_{\boldsymbol{\theta}^{(k)}}\mu_{\boldsymbol{\theta}^{(k)}}(\boldsymbol{\theta}^{(k)}) - \Pi_{\boldsymbol{\theta}^{(k-1)}}\mu_{\boldsymbol{\theta}^{(k-1)}}(\boldsymbol{\theta}^{(k)})\right) \\
&\quad + \left(\Pi_{\boldsymbol{\theta}^{(k-1)}}\mu_{\boldsymbol{\theta}^{(k-1)}}(\boldsymbol{\theta}^{(k)}) - \Pi_{\boldsymbol{\theta}^{(k)}}\mu_{\boldsymbol{\theta}^{(k)}}(\boldsymbol{\beta}^{(k+1)})\right), \\
&= \mathrm{D3\text{-}1} + \mathrm{D3\text{-}2} + \mathrm{D3\text{-}3}.
\end{aligned}
$$

(i) $\mu_{\boldsymbol{\theta}^{(k)}}(\boldsymbol{\beta}^{(k+1)}) - \Pi_{\boldsymbol{\theta}^{(k)}}\mu_{\boldsymbol{\theta}^{(k)}}(\boldsymbol{\theta}^{(k)})$ forms a martingale difference sequence since

$$\mathbb{E}\left[\mu_{\boldsymbol{\theta}^{(k)}}(\boldsymbol{\beta}^{(k+1)}) - \Pi_{\boldsymbol{\theta}^{(k)}}\mu_{\boldsymbol{\theta}^{(k)}}(\boldsymbol{\theta}^{(k)})|\mathcal{F}_k\right] = 0. \tag{D3-1}$$

(ii) From (33) and (30) respectively, we deduce that

$$\|\Pi_{\boldsymbol{\theta}^{(k)}}\mu_{\boldsymbol{\theta}^{(k)}}(\boldsymbol{\theta}^{(k)}) - \Pi_{\boldsymbol{\theta}^{(k-1)}}\mu_{\boldsymbol{\theta}^{(k-1)}}(\boldsymbol{\theta}^{(k)})\| \leq C_4\|\boldsymbol{\theta}^{(k)} - \boldsymbol{\theta}^{(k-1)}\| \leq 2C_4 C_0$$

Thus there exists $\overline{C}_4 = 2C_4 C_0$ such that

$$2\omega^{(k+1)}\langle \boldsymbol{T}^{(k)}, \Pi_{\boldsymbol{\theta}^{(k)}}\mu_{\boldsymbol{\theta}^{(k)}}(\boldsymbol{\theta}^{(k)}) - \Pi_{\boldsymbol{\theta}^{(k-1)}}\mu_{\boldsymbol{\theta}^{(k-1)}}(\boldsymbol{\theta}^{(k)})\rangle \leq \omega^{(k+1)}\overline{C}_4. \tag{D3-2}$$

(iii) D3-3 can be further decomposed to D3-3a and D3-3b

$$
\begin{aligned}
&\langle \boldsymbol{T}^{(k)}, \Pi_{\boldsymbol{\theta}^{(k-1)}}\mu_{\boldsymbol{\theta}^{(k-1)}}(\boldsymbol{\theta}^{(k)}) - \Pi_{\boldsymbol{\theta}^{(k)}}\mu_{\boldsymbol{\theta}^{(k)}}(\boldsymbol{\beta}^{(k+1)})\rangle \\
&= \left(\langle \boldsymbol{T}^{(k)}, \Pi_{\boldsymbol{\theta}^{(k-1)}}\mu_{\boldsymbol{\theta}^{(k-1)}}(\boldsymbol{\theta}^{(k)})\rangle - \langle \boldsymbol{T}^{(k+1)}, \Pi_{\boldsymbol{\theta}^{(k)}}\mu_{\boldsymbol{\theta}^{(k)}}(\boldsymbol{\beta}^{(k+1)})\rangle\right) \\
&\quad + \left(\langle \boldsymbol{T}^{(k+1)}, \Pi_{\boldsymbol{\theta}^{(k)}}\mu_{\boldsymbol{\theta}^{(k)}}(\boldsymbol{\beta}^{(k+1)})\rangle - \langle \boldsymbol{T}_k, \Pi_{\boldsymbol{\theta}^{(k)}}\mu_{\boldsymbol{\theta}^{(k)}}(\boldsymbol{\beta}^{(k+1)})\rangle\right) \\
&= (\boldsymbol{z}^{(k)} - \boldsymbol{z}^{(k+1)}) + \langle \boldsymbol{T}^{(k+1)} - \boldsymbol{T}^{(k)}, \Pi_{\boldsymbol{\theta}^{(k)}}\mu_{\boldsymbol{\theta}^{(k)}}(\boldsymbol{\beta}^{(k+1)})\rangle \\
&= \mathrm{D3\text{-}3a} + \mathrm{D3\text{-}3b},
\end{aligned}
$$

where $\boldsymbol{z}^{(k)} = \langle \boldsymbol{T}^{(k)}, \Pi_{\boldsymbol{\theta}^{(k-1)}} \mu_{\boldsymbol{\theta}^{(k-1)}}(\boldsymbol{\theta}^{(k)}) \rangle$ with a constant $\overline{C}_3 = 4C_0C_3$ which satisfies that

$$
\begin{aligned}
2\omega^{(k+1)} &\langle \boldsymbol{T}^{(k+1)} - \boldsymbol{T}^{(k)}, \Pi_{\boldsymbol{\theta}^{(k)}} \mu_{\boldsymbol{\theta}^{(k)}}(\boldsymbol{\beta}^{(k+1)}) \rangle \\
&= 2\omega^{(k+1)} \langle \boldsymbol{\theta}^{(k+1)} - \boldsymbol{\theta}^{(k)}, \Pi_{\boldsymbol{\theta}^{(k)}} \mu_{\boldsymbol{\theta}^{(k)}}(\boldsymbol{\beta}^{(k+1)}) \rangle \\
&\leq 2\omega^{(k+1)} \|\boldsymbol{\theta}^{(k+1)} - \boldsymbol{\theta}^{(k)}\| \cdot \|\Pi_{\boldsymbol{\theta}^{(k)}} \mu_{\boldsymbol{\theta}^{(k)}}(\boldsymbol{\beta}^{(k+1)})\| \\
&\leq 4\omega^{(k+1)} C_0 C_3 = \overline{C}_3 \omega^{(k+1)}
\end{aligned}
\tag{D3-3b}
$$

Finally, add all the items D1, D2 and D3 together, for some $\overline{C}_1 = \overline{C}_3 + \overline{C}_4$, we have

$$
\begin{aligned}
\mathbb{E}\left[\|\boldsymbol{T}^{(k+1)}\|^2\right] \leq {} & (1 - 2\omega^{(k+1)}\delta + \omega^{(k+1)^2} C_2) \mathbb{E}\left[\|\boldsymbol{T}_k\|^2\right] \\
& + \omega^{(k+1)^2} C_2 + \omega^{(k+1)} \overline{C}_1 + 2\omega^{(k+1)} \mathbb{E}[z^{(k)} - z^{(k+1)}].
\end{aligned}
$$

Moreover, from (33) and (30), there exists a constant $C_5$ such that

$$
\mathbb{E}[|\boldsymbol{z}^{(k)}|] \leq C_5.
\tag{38}
$$

Lemma 3 is an extension of Lemma 26 (page 248) from Benveniste et al. (1990).

**Lemma 3.** *Let $\{u^{(k)}\}_{k \geq k_0}$ as a sequence of real numbers such that for all $k \geq k_0$, some suitable constants $\overline{C}_1$ and $C_2$*

$$
u^{(k+1)} \geq u^{(k)} \left(1 - 2\omega^{(k+1)}\delta + \omega^{(k+1)^2} C_2\right) + \omega^{(k+1)^2} C_2 + \omega^{(k+1)} \overline{C}_1,
\tag{39}
$$

*and assume there exists such $k_0$ that*

$$
\mathbb{E}\left[\|\boldsymbol{T}^{(k_0)}\|^2\right] \leq u^{(k_0)}.
\tag{40}
$$

*Then for all $k > k_0$, we have*

$$
\mathbb{E}\left[\|\boldsymbol{T}^{(k)}\|^2\right] \leq u^{(k)} + \sum_{j=k_0+1}^{k} \Lambda_j^k (z^{(j-1)} - z^{(j)}).
$$

**Proof of Theorem 1 (Continued).** From Lemma 2, we can choose $\lambda_0$ and $k_0$ which satisfy the conditions (39) and (40)

$$
\mathbb{E}[\|\boldsymbol{T}^{(k_0)}\|^2] \leq u^{(k_0)} = \lambda_0 \omega^{(k_0)}.
$$

From Lemma 3, it follows that for all $k > k_0$

$$
\mathbb{E}\left[\|\boldsymbol{T}_k\|^2\right] \leq u^{(k)} + \mathbb{E}\left[\sum_{j=k_0+1}^{k} \Lambda_j^k \left(\boldsymbol{z}^{(j-1)} - \boldsymbol{z}^{(j)}\right)\right].
\tag{41}
$$

From (38) and the increasing property of $\Lambda_j^k$ in Lemma 1, we have

$$
\begin{aligned}
\mathbb{E}\left[\left\|\sum_{j=k_0+1}^{k} \Lambda_j^k \left(\boldsymbol{z}^{(j-1)} - \boldsymbol{z}^{(j)}\right)\right\|\right] &= \mathbb{E}\left[\left\|\sum_{j=k_0+1}^{k-1} (\Lambda_{j+1}^k - \Lambda_j^k)\boldsymbol{z}^{(j)} - 2\omega^{(k)}\boldsymbol{z}^{(k)} + \Lambda_{k_0+1}^k \boldsymbol{z}^{(k_0)}\right\|\right] \\
&\leq \sum_{j=k_0+1}^{k-1} (\Lambda_{j+1}^k - \Lambda_j^k) C_5 + \mathbb{E}[|2\omega^{(k)}\boldsymbol{z}^{(k)}|] + \Lambda_k^k C_5 \\
&\leq (\Lambda_k^k - \Lambda_{k_0}^k) C_5 + \Lambda_k^k C_5 + \Lambda_k^k C_5 \\
&\leq 3\Lambda_k^k C_5 = 6C_5\omega^{(k)}.
\end{aligned}
\tag{42}
$$

Therefore, given the sequence $u^{(k)} = \lambda_0 \omega^{(k)}$ that satisfies conditions (39), (40) and Lemma 3, for any $k > k_0$, from (41) and (42), we have

$$
\mathbb{E}[\|\boldsymbol{T}^{(k)}\|^2] \leq u^{(k)} + 3C_5 \Lambda_k^k = (\lambda_0 + 6C_5) \omega^{(k)} = \lambda \omega^{(k)},
$$

where $\lambda = \lambda_0 + 6C_5$. $\qquad \square$

## B.2 CONVERGENCE OF SAMPLES

In addition to the previous assumptions, we make one more assumption on the stochastic gradients to guarantee that the samples converge to the posterior conditioned on the optimal latent variables:

**Assumption 5** (Gradient Unbiasedness and Smoothness). *For all $\boldsymbol{\beta} \in \boldsymbol{B}$ and $\boldsymbol{\theta} \in \Theta$, the mini-batch of data $\mathcal{B}$, the stochastic noise $\boldsymbol{\xi}$, which comes from $\tilde{\nabla} L(\boldsymbol{\beta}, \boldsymbol{\theta}) - \nabla L(\boldsymbol{\beta}, \boldsymbol{\theta})$, is a white noise and independent with each other. In addition, there exists a constant $l \geq 2$ such that the following conditions hold:*

$$\mathbb{E}_{\mathcal{B}}[\boldsymbol{\xi}] = 0 \text{ and } \mathbb{E}_{\mathcal{B}}[\boldsymbol{\xi}^l] < \infty. \tag{43}$$

*For all $\boldsymbol{\theta}, \boldsymbol{\theta}' \in \Theta$, there exists a constant $M > 0$ such that the gradient is M-smooth:*

$$\nabla L(\boldsymbol{\beta}, \boldsymbol{\theta}) - \nabla L(\boldsymbol{\beta}, \boldsymbol{\theta}') \leq M \|\boldsymbol{\theta} - \boldsymbol{\theta}'\| \tag{44}$$

**Corollary 1.** For all $\alpha \in (0, 1]$, under assumptions 1-5, the distribution of $\boldsymbol{\beta}^{(k)}$ converges weakly to the invariant distribution $e^{L(\boldsymbol{\beta}, \boldsymbol{\theta}^*)}$ as $\epsilon \to 0$.

*Proof.* The proof framework follows from section 4 of Sato and Nakagawa (2014). In the context of stochastic noise $\boldsymbol{\xi}^{(k)}$, we ignore the subscript of $\epsilon$ and only consider the case of $\tau = 1$. Since $\boldsymbol{\theta}^{(k)}$ converges to $\boldsymbol{\theta}^*$ in SGLD-SA and the gradient is M-smooth (44), we transform the stochastic gradient from $\nabla \tilde{L}(\boldsymbol{\beta}^{(k)}, \boldsymbol{\theta}^{(k)})$ to $\nabla L(\boldsymbol{\beta}^{(k)}, \boldsymbol{\theta}^*) + \boldsymbol{\xi}^{(k)} + \mathcal{O}(\epsilon^\alpha)$, therefore Eq.(24) can be written as

$$\boldsymbol{\beta}^{(k+1)} = \boldsymbol{\beta}^{(k)} + \epsilon \left( \nabla L(\boldsymbol{\beta}^{(k)}, \boldsymbol{\theta}^*) + \boldsymbol{\xi}^{(k)} + \mathcal{O}(\epsilon^\alpha) \right) + \sqrt{2\epsilon} \boldsymbol{\eta}^{(k)}, \text{ where } \boldsymbol{\eta}^{(k)} \sim \mathcal{N}(0, \boldsymbol{I}). \tag{45}$$

Using Eq.(43), the characteristic function of $\epsilon \boldsymbol{\xi}^{(k)} + \mathcal{O}(\epsilon^{\alpha+1})$ is

$$\mathbb{E}\left[ \exp\left( is(\epsilon \boldsymbol{\xi}^{(k)} + \mathcal{O}(\epsilon^{\alpha+1})) \right) \right] = \mathbb{E}\left[ \sum_{l=0}^{\infty} \frac{1}{l!} \left( is(\epsilon \boldsymbol{\xi}^{(k)} + \mathcal{O}(\epsilon^{\alpha+1})) \right)^l \right] = 1 + \mathcal{O}(\epsilon^{\alpha+1}). \tag{46}$$

Then the characteristic function of $\sqrt{2\epsilon} \boldsymbol{\eta}^{(k)}$ is $\exp(-\epsilon s^2)$.

Rewrite $\boldsymbol{\beta}^{(k+1)}$ as $\boldsymbol{\beta}^{(k+\epsilon)}$, the characteristic function of $\boldsymbol{\beta}^{(t+\epsilon)}$ is the characteristic function of $\boldsymbol{\beta}^{(k)} + \epsilon \left( \partial_{\boldsymbol{\beta}} L(\boldsymbol{\beta}^{(k)}, \boldsymbol{\theta}^*) + \boldsymbol{\xi}^{(k)} + \mathcal{O}(\epsilon^\alpha) \right) + \sqrt{2\epsilon} \boldsymbol{\eta}^{(k)}$, which is

$$\begin{aligned}
\phi_{t+\epsilon}(s) &= \int \exp(is\boldsymbol{\beta} + is\epsilon \nabla L(\boldsymbol{\beta}, \boldsymbol{\theta}^*) - \epsilon s^2) \left( 1 + \mathcal{O}(\epsilon^{\alpha+1}) \right) q(k, \boldsymbol{\beta}) d\boldsymbol{\beta} \\
&= \int \exp(is\boldsymbol{\beta} + is\epsilon \nabla L(\boldsymbol{\beta}, \boldsymbol{\theta}^*) - \epsilon s^2) q(k, \boldsymbol{\beta}) d\boldsymbol{\beta} + \mathcal{O}(\epsilon^{\alpha+1}).
\end{aligned} \tag{47}$$

With the fact $\exp(x) = 1 + x + \mathcal{O}(x^2)$, we can get

$$\begin{aligned}
&\phi^{(k+\epsilon)}(s) - \phi^{(k)}(s) \\
&= \int \exp(is\boldsymbol{\beta}) \left( \exp(is\epsilon \nabla L(\boldsymbol{\beta}, \boldsymbol{\theta}^*) - \epsilon s^2) - 1 \right) q(k, \boldsymbol{\beta}) d\boldsymbol{\beta} + \mathcal{O}(\epsilon^{\alpha+1}) \\
&= \int \exp(is\boldsymbol{\beta}) \left( is\epsilon \nabla L(\boldsymbol{\beta}, \boldsymbol{\theta}^*) - \epsilon s^2 + \mathcal{O}(\epsilon^2) \right) q(k, \boldsymbol{\beta}) d\boldsymbol{\beta} + \mathcal{O}(\epsilon^{\alpha+1}) \\
&= \int \exp(is\boldsymbol{\beta}) \left( is\epsilon \nabla L(\boldsymbol{\beta}, \boldsymbol{\theta}^*) - \epsilon s^2 \right) q(k, \boldsymbol{\beta}) d\boldsymbol{\beta} + \mathcal{O}(\epsilon^{\alpha+1}).
\end{aligned} \tag{48}$$

Therefore,

$$\begin{aligned}
\frac{\phi^{(k+\epsilon)}(s) - \phi^{(k)}(s)}{\epsilon} &= -is \int \exp(is\boldsymbol{\beta}) \nabla(-L(\boldsymbol{\beta}, \boldsymbol{\theta}^*)) q(k, \boldsymbol{\beta}) d\boldsymbol{\beta} \\
&\quad + (-is)^2 \int \exp(is\boldsymbol{\beta}) q(k, \boldsymbol{\beta}) d\boldsymbol{\beta} + \mathcal{O}(\epsilon^\alpha).
\end{aligned} \tag{49}$$

Table 3: Predictive errors in logistic regression based on a test set considering different $v_0$ and $\sigma$

| MAE / MSE | $v_0=10^{-2},\sigma=0.6$ | $v_0=10^{-1},\sigma=0.6$ | $v_0=10^{-2},\sigma=2$ | $v_0=10^{-3},\sigma=2$ |
|---|---|---|---|---|
| SGLD-SA | **0.182 / 0.0949** | **0.195 / 0.1039** | **0.146** / 0.1049 | **0.165 / 0.0890** |
| SGLD | 0.311 / 0.2786 | 0.304 / 0.2645 | 0.333 / 0.2977 | 0.331 / 0.3037 |
| ESM | 0.240 / 0.0982 | 0.454 / 0.2080 | 0.182 / **0.0882** | 0.172 / 0.1102 |

For any integrable function $f$, set F as the Fourier transform defined by

$$\mathrm{F}[f(x)](s) = \frac{1}{\sqrt{2\pi}} \int f(x)\exp(isx)dx. \tag{50}$$

The inverse Fourier transform of $\mathrm{F}[f(x)]$ and the $l$-th order derivatives of $f^{(l)}(x)$ is

$$\mathrm{F}^{-1}[f(x)](s) = f(x) = \frac{1}{\sqrt{2\pi}} \int \mathrm{F}[f(x)](s)\exp(-isx)dx,$$
$$\mathrm{F}(f^{(l)})(s) = (-is)^l(\mathrm{F}(f))(s). \tag{51}$$

Combine Eq.(49), Eq.(50) and Eq.(51), we arrive at the following simplified equation:

$$\frac{\phi^{(t+\epsilon)}(s) - \phi^{(k)}(s)}{\sqrt{2\pi}\epsilon}$$
$$= -is\mathrm{F}\partial_{\boldsymbol{\beta}}(-L(\boldsymbol{\beta},\boldsymbol{\theta}^*))q(k,\boldsymbol{\beta}) + (-is)^2\mathrm{F}q(k,\boldsymbol{\beta}) + \mathcal{O}(\epsilon^\alpha)$$
$$= \mathrm{F}\partial_{\boldsymbol{\beta}}\left(\partial_{\boldsymbol{\beta}}(-L(\boldsymbol{\beta},\boldsymbol{\theta}^*))q(k,\boldsymbol{\beta})\right) + \mathrm{F}\partial_{\boldsymbol{\beta}}^2 q(k,\boldsymbol{\beta}) + \mathcal{O}(\epsilon^\alpha). \tag{52}$$

Since $\mathrm{F}^{-1}[\phi^{(k)}(s)] = \sqrt{2\pi}q(k,\boldsymbol{\beta})$ and $\alpha \in (0,1]$,

$$\lim_{\epsilon\to 0}\frac{q(k+\epsilon,\boldsymbol{\beta}) - q(k,\boldsymbol{\beta})}{\epsilon}$$
$$= \lim_{\epsilon\to 0}\mathrm{F}^{-1}\left[\frac{\phi^{(k+\epsilon)}(s) - \phi^{(k)}(s)}{\sqrt{2\pi}\epsilon}\right]$$
$$= \lim_{\epsilon\to 0}\partial_{\boldsymbol{\beta}}(\partial_{\boldsymbol{\beta}}(-L(\boldsymbol{\beta},\boldsymbol{\theta}^*))q(k,\boldsymbol{\beta})) + \partial_{\boldsymbol{\beta}}^2 q(k,\boldsymbol{\beta}) + \mathcal{O}(\epsilon^\alpha)$$
$$= \partial_{\boldsymbol{\beta}}(\partial_{\boldsymbol{\beta}}(-L(\boldsymbol{\beta},\boldsymbol{\theta}^*))q(k,\boldsymbol{\beta})) + \partial_{\boldsymbol{\beta}}^2 q(k,\boldsymbol{\beta}). \tag{53}$$

Finally, we have proved that the distribution of $\boldsymbol{\beta}^{(k)}$ converges weakly to the invariant distribution $e^{L(\boldsymbol{\beta},\boldsymbol{\theta}^*)}$ as $\epsilon \to 0$. $\qquad\square$

## C  SIMULATION OF LOGISTIC REGRESSION

Now we conduct the experiments on binary logistic regression. The setup is similar as before, except $n$ is set to 500, $\Sigma_{i,j} = 0.3^{|i-j|}$ and $\boldsymbol{\eta} \sim \mathcal{N}(0, \boldsymbol{I}/2)$. We set the learning rates in SGLD-SA and SGLD to $0.01 \times k^{-\frac{1}{3}}$ and step size $\omega^{(k)}$ to $0.1 \times (k+1000)^{-\frac{3}{4}}$. The binary response values are simulated from **Bernoulli**$(p)$ where $p = 1/(1 + e^{-X\boldsymbol{\beta}-\boldsymbol{\eta}})$. Fig.4(a), Fig.4(b) and Fig.4(c) demonstrate the posterior distribution of SGLD-SA is significantly better than that of SGLD. As shown in Fig.4(f), SGLD-SA is the best method to regulate the over-fitting space and provides the most reasonable posterior mean. Table.3 illustrates the predictive power of SGLD-SA is overall better than the other methods and robust to different initializations. Fig.4(d) and Fig.4(e) show that the over-fitting problem of SGLD when $p > n$ in logistic regression and the algorithm fails to regulate the over-fitting space; We observe SGLD-SA is able to resist over-fitting and always yields reproducible results.

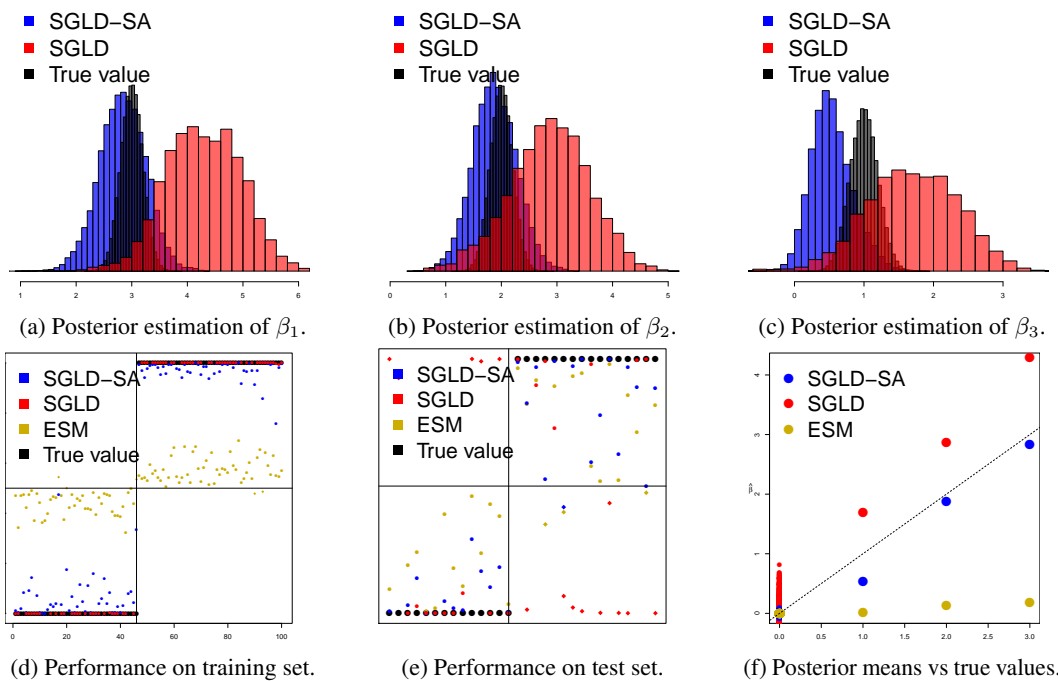

Figure 4: Logistic regression simulation when $v_0 = 0.1$ and $\sigma = 0.6$

## D EXPERIMENTAL SETUP

### D.1 NETWORK ARCHITECTURE

The first DNN we use is a standard 2-Conv-2-FC CNN: it has two convolutional layers with a $2 \times 2$ max pooling after each layer and two fully-connected layers. The filter size in the convolutional layers is $5 \times 5$ and the feature maps are set to be 32 and 64, respectively (Jarrett et al., 2009). The fully-connected layers (FC) have 200 hidden nodes and 10 outputs. We use the rectified linear unit (ReLU) as activation function between layers and employ a cross-entropy loss.

The second DNN is a 2-Conv-BN-3-FC CNN: it has two convolutional layers with a $2 \times 2$ max pooling after each layer and three fully-connected layers with batch normalization applied to the first FC layer. The filter size in the convolutional layers is $4 \times 4$ and the feature maps are both set to 64. We use $256 \times 64 \times 10$ fully-connected layers.

### D.2 TEMPERATURE STRATEGY

In practice, we observe a suitable temperature setting is helpful to improve the classification accuracy. For example, by setting $\tau = 100$ in the second DNN (see Appendix D.1) we obtain 99.70% on aMNIST. To account for the scale difference of weights in different layers, we apply different temperatures to different layers based on different standard deviations of the gradients in each layer and obtain the results in Tab. 2.

### D.3 DATA AUGMENTATION

The MNIST dataset is augmented by (1) randomCrop: randomly crop each image with size 28 and padding 4, (2) random rotation: randomly rotate each image by a degree in $[-15°, +15°]$, (3) normalization: normalize each image with empirical mean 0.1307 and standard deviation 0.3081.

The FMNIST dataset is augmented by (1) randomCrop: same as MNIST, (2) randomHorizontalFlip: randomly flip each image horizontally, (3) normalization: same as MNIST.

