# OpenReview forum: "Bayesian Deep Learning via Stochastic Gradient MCMC with a Stochastic Approximation Adaptation"
_ICLR.cc/2019/Conference_

### Official Review · AnonReviewer2 · 2018-10-31
**The proposed SGLD-SA algorithm with its convergence properties is interesting**

**Rating:** 6
**Confidence:** 5

**Review:**

* The proposed SGLD-SA algorithm, together with its convergence properties, is very interesting. The introduction of step size $w^{k}$ is very similar to the "convex combination rule" in (Zhang & Brand 2017) to guarantee convergence.

* It seems that this paper only introduced Bayesian inference in the output layers. It would be more interesting to have a complete Bayesian model for the full network including the inner and activation layers.

* This paper imposed spike-and-slab prior on the weight vector which can yield sparse connectivity. Similar ideas have been explored to compress the model size of deep networks (Lobacheva, Chirkova and Vetrov 2017; Louizos, Ullrich and Welling 2017 ). It would make this paper stronger to compare the sparsification and compression properties with the above work.

* In equation (11) there is a summation from $\beta_{p+1}$ to $\beta_{p+u}$. I wonder where this term comes from, as I thought $\beta$ is a vector of dimension $p$.

Reference:
Zhang, Ziming, and Matthew Brand. "Convergent block coordinate descent for training tikhonov regularized deep neural networks." Advances in Neural Information Processing Systems. 2017.

Lobacheva, Ekaterina, Nadezhda Chirkova, and Dmitry Vetrov. "Bayesian Sparsification of Recurrent Neural Networks." arXiv preprint arXiv:1708.00077 (2017).

Louizos, Christos, Karen Ullrich, and Max Welling. "Bayesian compression for deep learning." Advances in Neural Information Processing Systems. 2017.

---

### Official Review · AnonReviewer1 · 2018-11-02
**Interesting work, but in my view not substantial novelty and significance**

**Rating:** 4
**Confidence:** 2

**Review:**

TITLE
Bayesian deep learning via stochastic gradient mcmc with a stochastic approximation adaptation

REVIEW SUMMARY
Fairly well written paper on SG-MCMC type inference in neural networks with slab and spike priors. In my view, the originality and significance is limited.

PAPER SUMMARY
The paper develops a method for sampling/optimization of a Bayesian neural network with slab and spike priors on the weights.

QUALITY
I belive the contribution is technically sound (but I have not checked all equations or the proof of Theorem 1). The empirical evaluation is not unreasonable, but also not strongly convincing.

CLARITY
The paper is fairly well written, but grammar and use of English could be slightly improved (not so important).

ORIGINALITY
The paper builds on existing work on EM-type algorithms for slab and spike models and SG-MCMC for Bayesian inference in neural networks. The novelty of the contribution is limited: The main contribution is the combination of the two methods and some theoretical results. I am not able to judge if there is significant originality in the theoretical results (Theorem 1 + Corr 1+2) but if I am not mistaken it is more or less an application of a known result to this particular setting?

SIGNIFICANCE
While I think the proposed algorithm is reasonable and most likely useful in practice, I am not sure the contribution is substantial enough to gain large interest in the community.

FURTHER COMMENTS
Figure 2 (d+e) are in my view not so useful for assessing the training/test performance, but I am not even completely sure what the figures shows, as there are no axis labels. I would prefer some results on the loss, perhaps averaged over multiple data sets.

---

### Official Review · AnonReviewer3 · 2018-11-02
**Unclear benefits of SG-MCMC with SA and the experiments are not sufficiently convincing**

**Rating:** 5
**Confidence:** 4

**Review:**

The authors describe a new method of posterior sampling with latent variables based on SG-MCMC and stochastic approximation (SA). The new method uses a spike and slab prior on the weights of the deep neural networks to encourage sparsity. Experiments on toy regressions, classification and adversarial attacks demonstrate the superiority over SG-MCMC and EMSV.

Compared to the previous work EMSV (ESM), the novelty of SG-MCMC-SA is replacing the MAP in EMSV by SG-MCMC with stochastic approximation to alleviate the local trap problem in DNNs. However, I did not see why SG-MCMC with SA can achieve this goal. It is known that SG-MCMC methods tend to get trapped in a local optimal [1]. How did SA solve this problem? Besides, it is unclear to me where Eq. 17 uses stochastic approximation. The authors need to explain more about stochastic approximation for the readers who are not familiar with this method.

Empirical results on a synthetic example, MNIST and FMNIST show that SG-MCMC-SA outperforms the previous methods. However, the improvements of the proposed method are marginal. MNIST and FMNIST are small and easy datasets and it is very hard to tell the effectiveness of SG-MCMC-SA. It would be more convincing to show the empirical results on other datasets, e.g. CIFAR, using some larger architectures. The comparison would be more significant in that case.

[1]. Zhang, Yizhe, et al. "Stochastic Gradient Monomial Gamma Sampler." arXiv preprint arXiv:1706.01498 (2017).

---

### Meta-Review · Area_Chair1 · 2018-12-12
**Good Bayesian approach to deep networks with spike-and-slab prior but with limited originality and lack of experiment support**

**Confidence:** 3
**Recommendation:** Reject

**Metareview:**

This paper proposes a Bayesian alternative to dropout for deep networks by extending the EM-based variable selection method with SG-MCMC for sampling weights and stochastic approximation for tuning hyper-parameters. The method is well presented with a clear motivation. The combination of SMVS, SG-MCMC, and SA as a mixed optimization-sampling approach is technically sound.

The main concern raised by the readers is the limited originality. SG-MCMC has been studied extensively for Bayesian deep networks and applying the spike-and-slab prior as an alternative to dropout is a straightforward idea. The main contribution of the paper appears to be extending EMVS to deep net with commonly used sampling techniques for Bayesian networks.

Another concern is the lack of experimental justification for the advantage of the proposed method. While the authors promise to include more experiment results in the camera-ready version, it requires a considerable amount of effort and the decision unfortunately has to be made based on the current revision.